# Cryo-EM structure of ex vivo fibrils associated with extreme AA amyloidosis prevalence in a cat shelter

Tim Schulte[1,13], Antonio Chaves-Sanjuan [2,3,13], Giulia Mazzini [4,5], Valentina Speranzini [2], Francesca Lavatelli [4], Filippo Ferri[6], Carlo Palizzotto[6], Maria Mazza[7], Paolo Milani[4,5], Mario Nuvolone [4,5], Anne-Cathrine Vogt[8,9], Monique Vogel[8,9], Giovanni Palladini[4,5], Giampaolo Merlini [4,5], Martino Bolognesi [2,3], Silvia Ferro [10], Eric Zini[6,11,12] & Stefano Ricagno [1,2] ✉

AA amyloidosis is a systemic disease characterized by deposition of misfolded serum amyloid A protein (SAA) into cross-β amyloid in multiple organs in humans and animals. AA amyloidosis occurs at high SAA serum levels during chronic inflammation. Prion-like transmission was reported as possible cause of extreme AA amyloidosis prevalence in captive animals, e.g. 70% in cheetah and 57–73% in domestic short hair (DSH) cats kept in zoos and shelters, respectively. Herein, we present the 3.3 Å cryo-EM structure of AA amyloid extracted *post-mortem* from the kidney of a DSH cat with renal failure, deceased in a shelter with extreme disease prevalence. The structure reveals a cross-β architecture assembled from two 76-residue long proto-filaments. Despite >70% sequence homology to mouse and human SAA, the cat SAA variant adopts a distinct amyloid fold. Inclusion of an eight-residue insert unique to feline SAA contributes to increased amyloid stability. The presented feline AA amyloid structure is fully compatible with the 99% identical amino acid sequence of amyloid fragments of captive cheetah.

Amyloidosis is associated with the deposition of proteinaceous amorphous structures in the extracellular space of tissue and organs in humans and animals[1]. Amyloids in biopsies are histologically revealed by apple-green birefringence under polarized light after Congo Red staining[1,2]. Each of more than 50 disease-causing amyloidogenic proteins is associated with a specific disease type and organ distribution[1,3,4]. Immunohistochemistry and mass spectrometry-based determination of amyloid type is vital for effective treatment[2,5,6]. Single-particle cryo-EM has recently been applied to classify human brain amyloidoses (tauopathies) based on fibril structures, potentially impacting future diagnosis and treatment of these devastating neurodegenerative diseases[7]. Specifically, AA amyloidosis represents a

[1]Institute of Molecular and Translational Cardiology, IRCCS Policlinico San Donato, 20097 Milan, Italy. [2]Department of Biosciences, Università degli Studi di Milano, Milan, Italy. [3]Pediatric Research Center Fondazione R.E. Invernizzi and NOLIMITS Center, Università degli Studi di Milano, Milan, Italy. [4]Department of Molecular Medicine, University of Pavia, Pavia, Italy. [5]Amyloidosis Research and Treatment Center, Fondazione IRCCS Policlinico San Matteo, Pavia, Italy. [6]AniCura Istituto Veterinario Novara, Strada Provinciale 9, 28060 Granozzo con Monticello, Novara, Italy. [7]Istituto Zooprofilattico Sperimentale del Piemonte Liguria e Valle d'Aosta, S.C. Diagnostica Specialistica, Via Bologna 148, 10154 Torino, Italy. [8]Department for BioMedical Research (DBMR), University of Bern, 3008 Bern, Switzerland. [9]Department of Rheumatology and Immunology, University Hospital Bern, 3010 Bern, Switzerland. [10]Department of Comparative Biomedicine and Food Sciences, University of Padova, viale dell'Università 16, 35020 Legnaro, Padua, Italy. [11]Department of Animal Medicine, Production and Health, University of Padua, viale dell'Università 16, 35020 Legnaro, Padua, Italy. [12]Clinic for Small Animal Internal Medicine, Vetsuisse Faculty, University of Zurich, Winterthurerstrasse 260, 8057 Zurich, Switzerland. [13]These authors contributed equally: Tim Schulte, Antonio Chaves-Sanjuan. ✉e-mail: stefano.ricagno@unimi.it

systemic disease characterized by the deposition of misfolded serum amyloid A protein (SAA) in multiple organs[2,8]. This 12–14 kDa remarkably conserved apo-lipoprotein acts as molecular mop to remove membrane debris and lipid degradation products from inflammatory injury sites[9–12]. As part of the host innate response to inflammation, acute-phase SAA is secreted by the liver to increase serum levels up to 1000-fold[2,9,10,13–15]. A minor fraction of SAA adopts an α-helical bundle structure that delivers retinol to intestinal myeloid cells, including macrophages, to promote adaptive immunity[16,17]. The vast majority of SAA with a more disordered and enigmatic structure is bound in high-density lipoprotein (HDL), likely contributing to cholesterol homeostasis in macrophages[2,9,15,18,19]. Chronic inflammation leads to sustained increased SAA concentrations, to an extent that macrophages reportedly fail to prevent proteolysis-resistant oligomers during lysosomal degradation[20,21]. Low pH in lysosomes may favor transition of SAA into highly ordered almost indestructible amyloid[20–23]. Final assembly into massive AA amyloid deposits physically distorts and damages organs, in human patients often diagnosed as kidney-related glomerular proteinuria[2,24,25]. A comparison of cryo-electron microscopy (EM) structures of ex vivo AA amyloid from diseased organs of a human patient and an experimental mouse model revealed the characteristic cross-β architecture of amyloid, but distinct structures despite 76% sequence identity[26]. The two structures were added to a growing amyloid structure database exhibiting more diverse folds than originally anticipated[23,27]. Proteins of identical sequence may adopt different cross-β structures, that are defined in vitro by test tube conditions and ex vivo by tissue origin and disease type[3,7,26–30]. While amyloid polymorphs of the same protein are energetically similar, disease-associated amyloids appear more stable than functional reversible amyloid[31]. Due to the conserved amyloidogenic nature of SAA, domestic animals develop systemic amyloidosis similarly to humans[2,32,33]. Among domestic short hair (DSH) cats, Siamese and Abyssinian breeds were reported as particularly prone to amyloidosis due to a familial predisposition[34–38]. Strikingly, the close-to-extinct captive cheetah, from whose lineage DSH cat ancestors split about six million years ago, suffers from an extreme disease prevalence of 70%, reportedly facilitated by prion-like transmission[39–41]. The classical prion disease transmissible spongiform encephalopathies (TSE) is caused by conversion of the native cellular prion protein PrP^C into its amyloidogenic PrP^sc form, infectious as non-fibrillar ~600 kDa oligomer and larger amyloid fragments[42–46]. Protein aggregates spreading trans-cellularly through the human nervous system in Alzheimer's and Parkinson's disease have been referred to as prionoids, to acknowledge lacking evidence of horizontal transmission[47–49]. Importantly, SAA has been suggested as prion candidate, based on extreme disease prevalence in captive cheetah and experimentally accelerated AA amyloidosis in chronically inflamed animals after intravenous injection or oral administration of amyloid[2,48,50,51]. Our recent study has revealed

a prevalence of 57–73% among 80 domestic short hair (DSH) cats kept in shelters[52], in stark contrast to a very low prevalence (1–2%) in client-owned cats[53,54]. AA amyloid deposits were found in the kidney, liver, spleen, and even the heart[52]. SAA was present in cat bile, indicating a potential fecal-oral transmission route, as reported for captive cheetah[39,52].

Herein, we present the cryo-EM structure of fibrils extracted *post-mortem* from the diseased kidney of a DSH cat with systemic AA amyloidosis, deceased in a shelter with extreme disease prevalence. The structure exhibits the characteristic cross-β architecture of amyloid, but adopts a unique fold. The amino acid sequence of AA amyloid from a cheetah deceased in a zoo is fully compatible with the presented feline amyloid structure.

## Results and discussion
### AA amyloid extracted from the kidney of a DSH cat deceased with renal failure
During the last two months of a two-year stay in a shelter in Northern Italy, a female DSH cat became anorectic, developed jaundice and lost significant body weight. She was affected by chronic kidney and liver disease, and had no retroviral infections. Due to worsened renal failure, euthanasia was requested when the cat was six years old. Histology of the kidney revealed mild chronic multifocal interstitial nephritis and that of the liver showed severe diffuse hypotrophy/atrophy of the hepatocytes. Abundant, amorphous and eosinophilic material in the kidney, liver and spleen stained positive for Congo red and appeared green-apple birefringent under polarized light, consistent with amyloid (Fig. 1a and Supplementary Fig. 1a). We suspected AA amyloidosis, representing the most commonly observed type of amyloidosis in animals[2,32,33]. Antibodies raised against feline SAA-derived peptides co-stained Thioflavin S-positive tissue sections in the kidney, liver and spleen (Supplementary Fig. 1b), as reported previously for 34 additional cats with systemic AA amyloidosis in all three organs[52]. Peptides obtained from kidney-derived fibril extracts were analyzed by liquid chromatography with tandem mass spectrometry (LC-MS/MS) to identify residue numbers 19–111 of the feline SAA precursor (Q1T770), corresponding to residue numbers 1–93 of the mature protein lacking the signal peptide, as most abundant matches (Supplementary Table 1 and Supplementary Fig. 2). Based on negative stain electron microscopy (EM), revealing straight helical filaments with crossover distances in the 650–700 Å range (Fig. 1b), fibril extraction was optimized for collection of a high-resolution single-particle cryo-EM dataset.

### Cat AA amyloid is built from two identical 76-residue long protofilaments stabilized through staggered ionic lock and hydrophobic cluster interactions
Cryo-electron micrographs of vitrified AA amyloid extracts revealed a homogeneous population of straight fibrils (Fig. 2a) that were manually picked for standard helical reconstruction (Supplementary Fig. 3a)[55,56]. About 65k from initially 380k segments were refined with C2 symmetry, a left-handed twist angle of 1.3° and a helical rise of 4.9 Å to yield a final map with a nominal resolution of 3.3 Å, as estimated from half-map Fourier shell correlation curves (FSC) (Supplementary Fig. 3b). Reasonable map-model statistics as well as matching 2D class averages and map projections provide evidence of a physically valid model built into a consistently reconstructed map (Fig. 2a–d, Supplementary Fig. 4 and Supplementary Table 2). The fibril structure is composed of two identical proto-filaments, and exhibits the cross-β architecture characteristic of amyloid (Fig. 2). The polypeptide of each proto-filament comprises 11 β-strands between residue positions 19 and 94 and adopts an extended hairpin structure. Residues 95–111, identified by LC-MS/MS as component of the extracted AA amyloid, were not visible in the map. A central β-arch between residues Asp-50 and Arg-64 links two ~25 residue long meandering tails that stick

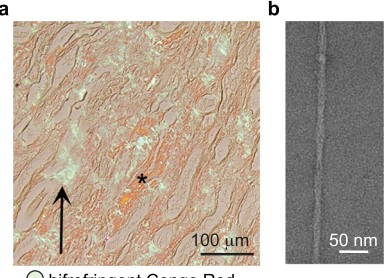

**Fig. 1 | AA amyloid extracted *post-mortem* from the kidney of a shelter cat deceased with renal failure. a** Abundant interstitial Congo Red-stained amyloid deposits appear orange-red (asterisks) with green-apple birefringence (arrows) under polarized light. **b** Micrograph of negative-stained fibril extracted from the kidney. See Supplementary Figs. 1 and 2.

○ bifrefringent Congo Red

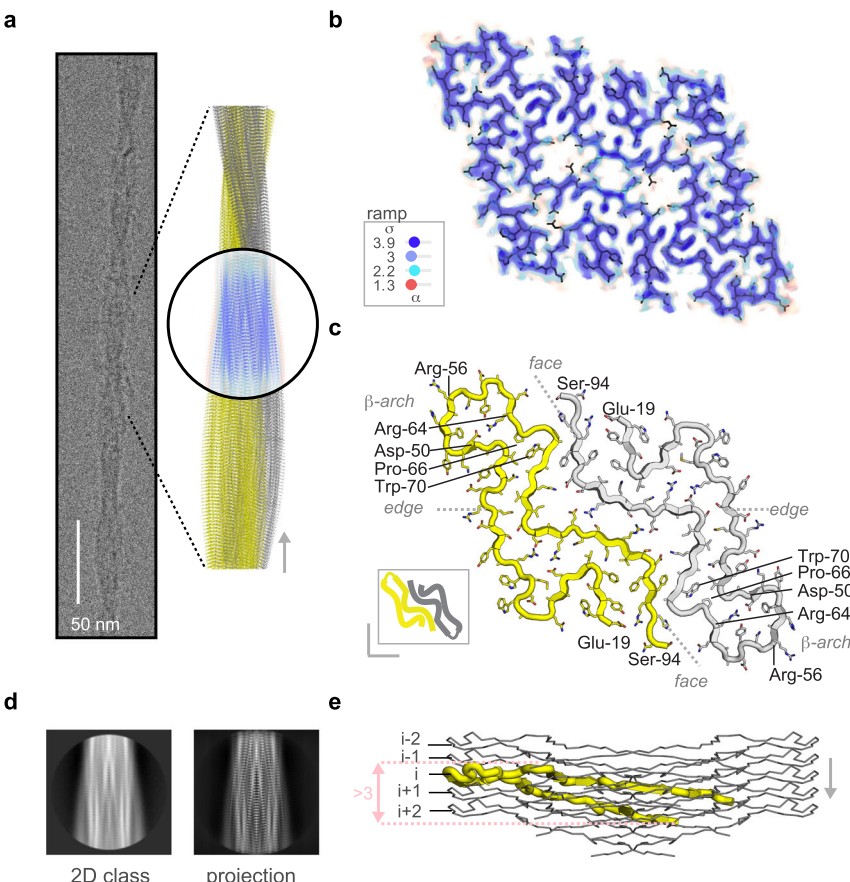

**Fig. 2 | The 3.3 Å resolution cryo-EM structure of cat AA amyloid extracted *post mortem* from the kidney. a** Cryo-EM image of a single straight fibril with a cross-over distance in the 650–700 Å range. The fibril model spans approximately an entire crossover length of 680 Å and was constructed using the deposited model (PDB: 7ZH7) composed of two proto-filaments (yellow and gray), each assembled by five chains. The map view was oriented to match the fibril orientation of the averaged 2D class and corresponding 2D projection of the reconstructed map. Compared to the color ramp in panel **b**, map values were set to 1.1, 2.0, 4.4 and 14.2 for better visualization of inter-chain distances. **b** Cross-sectional view of the map volume with contour levels according to the depicted σ-color scale. **c** The molecular model of two subunits within a single fibril layer is shown as cartoon with side chains in yellow and gray. N- and C-terminal positions of each chain and of the β-arch structure are indicated. Residues are numbered according to Uniprot entry Q1T770, corresponding to the feline SAA precursor before cleavage of the 18 residue long N-terminal signal peptide. A scheme in the lower left corner depicts the two chains in yellow and gray. **d** 2D class average corresponding to the orientation of the map shown in panel **a**. **e** Side-view of the deposited model comprising five subunits in each proto-filament. The N- and C-terminal tails are tilted by 10° and 15°, respectively, to the central β-arch that lies almost perpendicular to the long axis of the fibril. Cα-positions of Arg-56 were defined as rung levels (i, i ± 1 and i ± 2) along the long fibril axis. See Supplementary Figs. 3–5 for cryo-EM data processing workflow, quality indicators, additional 2D classes, projections, map views, local resolution map and *cis*-Proline.

together via side chain contacts. A noteworthy feature following the β-arch is an unusual backbone bulge adopted by the $P_{66}$GGAW$_{70}$ segment comprising Pro-66 modeled as *cis*-isomer (Supplementary Fig. 5a), in contrast to the *trans*-Proline residues in mouse and human AA amyloid (Supplementary Fig. 5b). To the best of our knowledge, this is the first example of a *cis*-Proline in amyloid. Isomerization of unfolded SAA may occur spontaneously, as observed in human dialysis-related amyloidosis of β2-microglobulin, but could also be catalyzed by isomerases[57–61]. In the assembled fibril, the N-terminal tails are surface-exposed at the edges, while the C-terminal tails are buried facing each other (Fig. 2c). Each polypeptide deviates from planarity traversing more than three rung layers (Fig. 2e). While the β-arch lies almost perpendicular to the fibril axis, the exposed edge- and buried face-tails are tilted by 15° and 10°, respectively. At the intra-protomer interface (Fig. 3 and Supplementary Fig. 6), the edge-tail of rung layer (i) contacts the face-tails *(i-1)* to *(i + 2)*, creating four hydrophobic clusters, four ionic locks and additional H-bond interactions. On the other side, at the inter-protomer interface (Fig. 3 and Supplementary Fig. 6), the face-tail (i) contacts four rung layers of the adjacent proto-filament, creating two hydrophobic clusters, four ionic locks and two additional H-bond interactions. Such staggered interactions contribute to fibril stability, as described previously[27].

## The distinct fold of cat AA amyloid buries its unique eight-residue insert between the two proto-filaments

Although the amino acid sequences of the published human and mouse AA amyloid structures[26] are >70% identical to the cat structure, each fold is distinct (Fig. 4). Compared to the 54-residue short core of human AA amyloid, the mouse and cat structures are elongated by 14 and 22 residues, respectively. Each structure adopts a unique fold, exhibiting distinct arrangements of β-strands that vary slightly in number and lengths, despite high sequence identities (Fig. 4a). In each fibril, different parts of the sequences are exposed or buried. In cat AA amyloid, residues 19–49 comprising strands β1-β4 are exposed as part of the edge-tail, comprising two short segments that are partially buried in sharp turns. Longer buried segments are observed for the corresponding region in both human and mouse AA amyloid, but with different distributions. Despite these differences, a segment between residues 24 and 54 of human AA amyloid superposes well on the cat structure with a rmsd-value of 2.5 Å (Supplementary Fig. 7). The

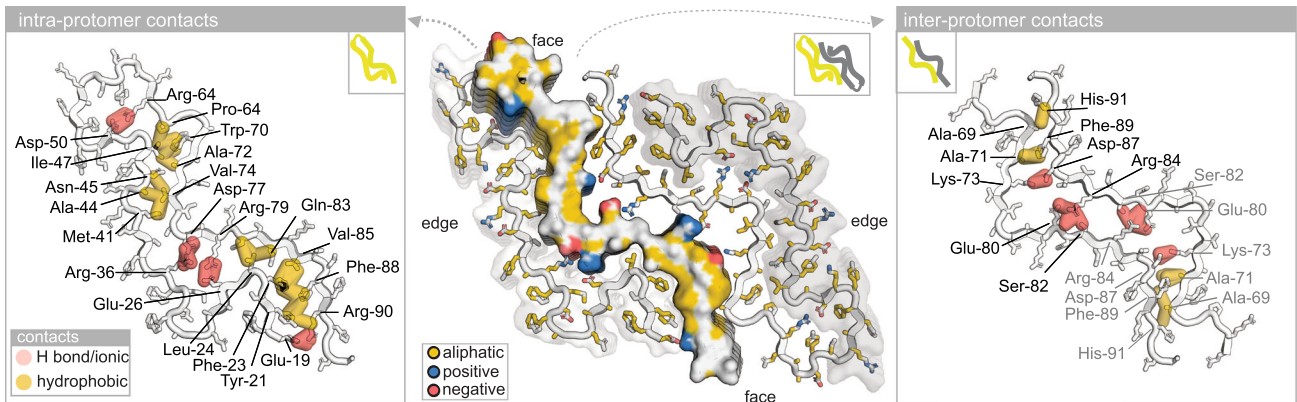

**Fig. 3 | Staggered ionic locks and hydrophobic clusters stabilize intra- and inter-protomer interfaces.** (Center) In the cross-sectional view the face of the left proto-filament is represented as molecular surface with aliphatic, positively and negatively charged side chain atoms in yellow, blue, and red, respectively. (Left, right) Side chain contacts at the intra- and inter-protomer interfaces are visualized in separate panels to the left and right, respectively. Hydrophobic and hydrogen (H) bond as well as ionic contacts are shown as yellow and pink semi-transparent heavy lines. The backbone and side chain atoms of the opposing strands are represented in mixed cartoon/stick format in white with black outlines. Residue contacts at the inter-protomer interface are related by a two-fold symmetry axis, which is highlighted by black and gray residue labels. See Supplementary Fig. 6 for molecular footprints to illustrate staggered contacts.

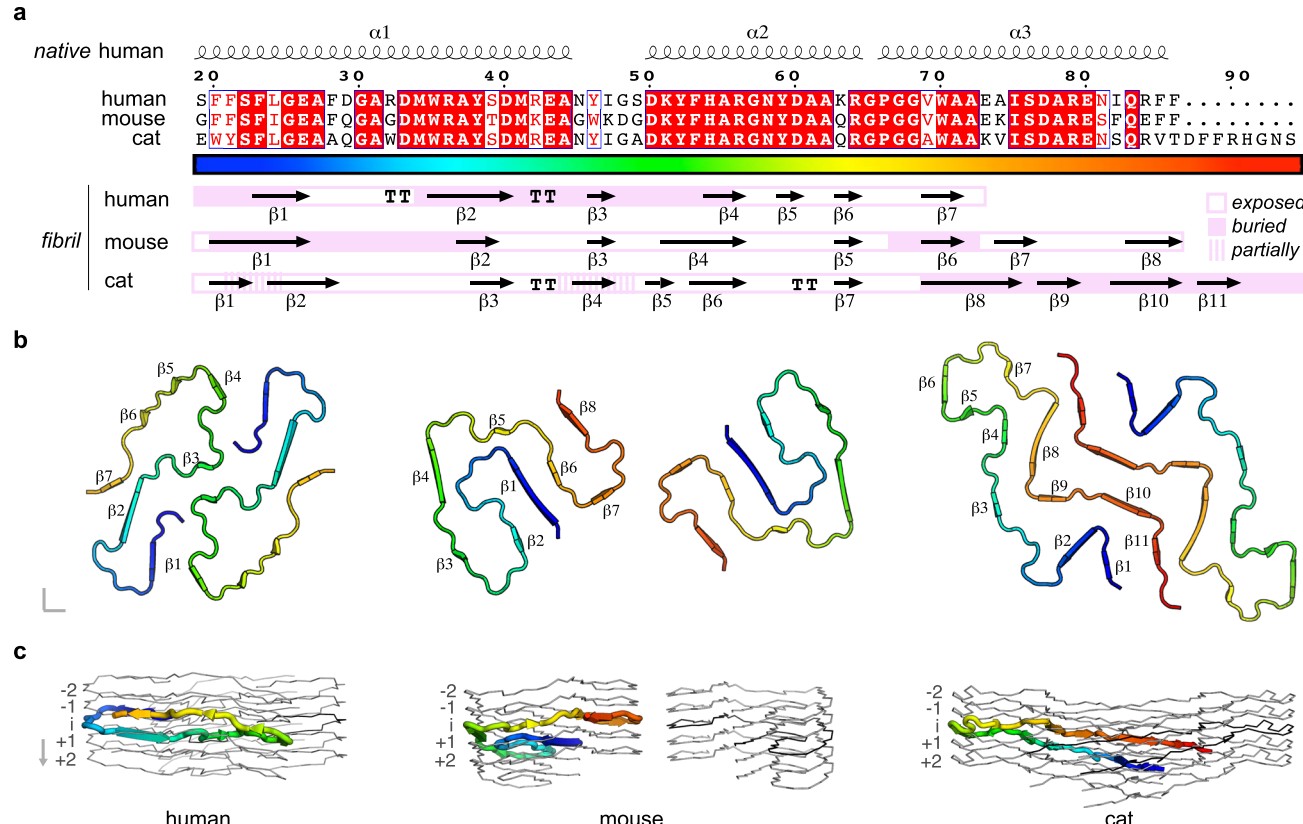

**Fig. 4 | Cat AA amyloid buries its unique eight-residue insert in an extended inter-protomer interface. a** The amino acid sequences of human, mouse and cat AA amyloid fibril cores (related Uniprot[90] entries of SAA precursors: P0DJI8, P05367 and Q1T770) are aligned and numbered according to the feline SAA precursor. The alignment was visualized using ESPript[91]. β-strands and strict β-turns are indicated by numbered β and non-numbered TT symbols, respectively. Strict sequence identity is indicated by a red box with white character, similarities within and across groups are indicated by red characters and blue frames, respectively. Secondary structure elements of the native human and three fibril structures are shown above and below the sequence alignment, respectively. Secondary structures were extracted from PDB[98] entries 4IP8, 6MST, 6DSO and 7ZH7, respectively. Buried, partially buried and exposed segments were assigned manually taking into account accessible surface areas and relative positioning of segments in the fibril. **b** Cross-section views of human, murine and cat fibrils illustrate the distinct molecular arrangements of strands and interfaces. Residues are colored according to the rainbow code in panel **a**. **c** Each chain in the human, mouse and cat fibril is not planar, but spans 11, 13.5 and 16.5 Å along the long fibril axis, corresponding to the crossings of about two (human) or three layers (mouse and cat). One chain per fibril is colored as in panel **b**, the other chains are shown as gray ribbons. See Supplementary Figs. 7–11 for analysis of shared structural elements, fibril surfaces, $\Delta G_{diss}$ and $\Delta G_{sol}$ analysis and native SAA structures.

concomitant observation of shared and distinct structural elements in sequence-homologous amyloid has been referred to as type-2 polymorphism[27]. The surface-exposed β-arch of cat AA amyloid, comprising residues 50 to 64, adopts more extended conformations in the two other structures. In human AA amyloid, residues 50–55 are buried, followed by an exposed C-terminal segment. In mouse AA amyloid, residues 50–64 are exposed, while residues 65–86 adopt a U-shaped structure that is, except for residues 66–72, largely exposed and in loose contact with the other protomer. A non-conserved sequence insertion at position 86 of the precursor protein sequence sets apart the cat from mouse and human SAA variants[10]. In the cat fibril, the insert constitutes a part of the buried tail at the inter-protomer interface.

### Inclusion of the eight-residue insert in cat AA amyloid increases mass, buried surface area and stability of fibril core

The described differences of the protein sequences and amyloid folds yield unique fibril architectures (Fig. 4, Supplementary Figs. 8 and 9). Structure-based solvation free energy ($\Delta G_{sol}$) calculations[31] reveal an estimated stability of −42 kcal/mol for a single cat AA amyloid molecule, similarly to mouse, but 5–10 kcal/mol more stable than human AA amyloid (Supplementary Fig. 10a). Typically, $\Delta G_{sol}$-values of disease-related amyloids range between −25 and −62 kcal/mole[31]. As an alternative estimate of fibril stability, we calculated the dissociation energy cost ($\Delta G_{diss}$)[62] of a single molecule from its respective fibril end (Supplementary Fig. 10b), which is about 4 and 8 kcal/mole higher for cat relative to mouse and human AA amyloid, respectively. The higher dissociation cost is mainly caused by the eight-residue insert that increases fibril core mass and enlarges the buried surface area by ~2000 Å². Additionally, the insert might destabilize feline native SAA (Supplementary Fig. 11) to increase levels of misfolded SAA available for fibrillation.

Remarkably, the amino acid sequence of AA amyloid extracted *post-mortem* from the diseased liver of the distantly related captive cheetah is 99% identical to the sequence of the extracted cat fibril, also comprising the eight-residue insert (Fig. 5). While cheetah AA amyloid may adopt an identical or a distinct fold, we consider the only N93S substitution in cheetah fully compatible with the herein presented structure. Captive cheetah in zoos suffer from a similarly high AA-amyloidosis prevalence of 70%, reportedly facilitated by a prion-like disease transmission[39,40,51]. Whether auto-catalytic, self-perpetuating and self-propagating amyloids are defined as prions, depends primarily on their capacity to undergo the full transmission cycle of an infective agent[44,45,63]. AA amyloid extracted from the liver and feces of cheetah was demonstrated to infect inflamed mice with elevated SAA precursor levels[39]. We may speculate that higher mass and increased structural stability of AA amyloid, as reported here for the feline variant, could increase its environmental persistence and resistance to clearance in the host. If fecal-oral transmission becomes accessible in

crowded populations of chronically inflamed felids, prion-like disease outbreaks might be recognized by extreme prevalence as reported for cat and cheetah in shelters and zoos[39,40,51,52].

In summary, here we report the 3.3 Å cryo-EM structure of feline fibrils associated with extreme AA amyloidosis prevalence in a cat shelter. The structure is unique in representing the first ex vivo structure of a spontaneously occurring amyloid obtained from an animal kept in a man-made habitat. Remarkably, amyloid extracted from captive cheetah with reported prion-like transmission is almost identical in sequence. Inclusion of the eight-residue insert unique to feline AA amyloid increases mass and stability of its fibril core, which could facilitate fecal-oral transmission.

## Methods
### Diagnosis of AA amyloidosis
**Cats and clinical data.** Amyloid was extracted from cat AAG17 which had been enrolled prospectively together with 79 other cats for our related study[52]. All animals were handled according to protocols approved by the Swiss Federal Veterinary Office (Application #30369). Kidney-derived fibril extracts were obtained from a sterilized female DSH cat with ID code AF755, which was euthanized at 6 years of age. The sterilized female DSH cat AG573 and castrated male DSH cat AI264, whose staining images are shown as controls (Supplementary Fig. 1c), was euthanized at 15 years of age and died at 8 years of age, respectively.

**Histology and immunofluorescence.** Full details were described previously[52]. In brief, organs were collected within 5 h from death, fixed in 10% formalin, and embedded in paraffin. After automatic sectioning, 4–5 μm-thick slices were stained with hematoxylin/eosin and Congo red and examined using standard and polarized light microscopy. After deparaffinization, amyloids and nuclei were stained with 1% (w/V) Thioflavine S and DAPI, respectively. For immunofluorescence, serum was obtained by immunization of Balb/c mice with peptide-conjugated virus-like particle (VLPs), as previously described in detail[52,64]. Residues 41–50 (MREANYIGAD), 63–74 (QRGPGGAWAAKV) and 109–122 (EWGRSGKDPNHFRP) of Uniprot entry P19707 were selected as antigens. Serum specificity was assessed using ELISA. Tissue was stained using serum, goat anti-mouse monoclonal IgG conjugated to biotin (1030-08, SouthernBiotech, Birmingham, Alabama) and streptavidin conjugated to Alexa-546 (s11225, Molecular Probes, Eugene, OR, USA) at 1:10, 1:500 and 1:500 dilutions, respectively. Images were acquired with AxioImager A2 and AxioCam (Carl Zeiss, Jena, Germany).

**Fibril extraction.** After excision, non-fixed cat kidneys were stored frozen (−80 °C) until amyloids were extracted as described previously[65–67]. Briefly, 0.5 g of kidney tissue (from the pole region) were minced with a scalpel and washed in Tris calcium buffer (20 mM Tris, 140 mM NaCl, 2 mM CaCl₂, pH 8.0). The tissue was digested with

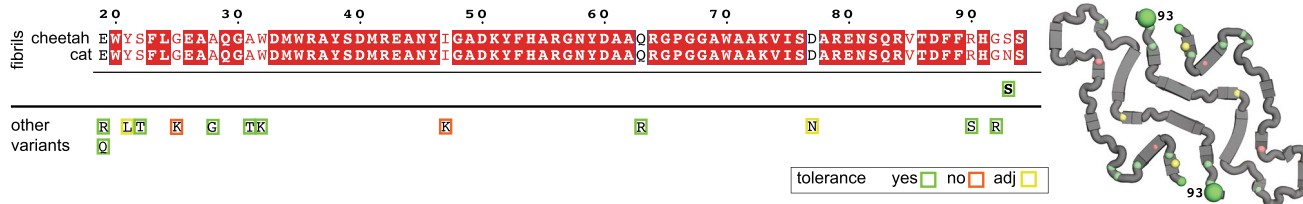

**Fig. 5 | Cheetah and cat AA amyloid are 99% identical.** Sequence alignment of extracted cat and cheetah amyloid (Q1T770 and B0M1H2) identified in this and a previous study[39]. Sequence conservation is based on a multiple sequence alignment[90,92] comprising ten cat and cheetah SAA variants (A0A2I2UCY9, A0A6J2AHC5, A0A337S9A8, A0A337SUS3, A0A6J2AJW0, M3WHE0, A0A5F5XYT5, A0A337SKP2). Single-residue substitutions are highlighted on sequence (left) and structure level (right). Substitution tolerance was estimated based on simple structural considerations, and colored in green (tolerated), yellow (tolerated with adjustments) and red (not tolerated). Cat AA amyloid is shown as gray cartoon. Large and small spheres highlight the positions of single-residue substitutions in fibrils and other SAA variants, respectively.

5 mg *Clostridium histolyticum* collagenase (Sigma Aldrich, Saint Louis, MO, USA) dissolved in 1 ml of Tris calcium buffer, followed by 10 cycles of homogenization in Tris EDTA buffer (1 ml, 20 mM Tris, 140 mM NaCl, 10 mM EDTA, pH 8.0). Supernatants were discarded after each homogenization step. The remaining pellet was then repeatedly homogenized in 0.5 ml ice-cold water, storing supernatants. Water extracts were analyzed by SDS-PAGE under denaturing and reducing conditions. The cryo-EM and proteomic analyses were performed on water extract #2 (Supplementary Fig. 2a). Fibril proteins were solubilized for 1 h at room temperature in 8 M urea/0.1 M DTT, and quantified using BioRad Protein assay (Bio-Rad, Hercules, CA, USA.

**LC-MS/MS**. 30 µg of solubilized and reduced protein was alkylated (150 mM iodoacetamide, 1 h, RT, dark), diluted with 100 mM $NH_4HCO_3$/5% acetonitrile (V/V) to a final urea concentration of 1.3 M, and digested with Trypsin (Sequence grade, Promega, Madison, WI, USA) at a 1:20 (w/w) ratio for 16 h at 37 °C. Peptides were purified using Pierce C18 Tips (Thermo Fisher Scientific) and analyzed by LC-MS/MS (Supplementary Table 1). Uniprot entries Q1T770, Q9XSG7, A0A337SKP2 and Q5XXU5, were identified as top hits from the *Felis catus* proteome.

### Structure of AA amyloid by single-particle cryo-EM
**Sample preparation and data collection.** A 4-µl droplet of fibrils sample was applied onto a C-flat thick 1.2/1.3 300 mesh Cu, previously glow-discharged for 30 s at 30 mA using a GloQube system (Quorum Technologies). The sample was blotted immediately and plunge-frozen in liquid ethane using a Vitrobot Mk IV (Thermo Fischer Scientific). A cryo-EM dataset of 2,652 movies was collected automatically on a Talos Arctica 200 kV (Thermo Fisher Scientific), equipped with a Falcon 3 direct electron detector operated in electron counting mode (Supplementary Table 2).

**Helical reconstruction.** Fibrils were picked manually from dose-weighted, motion- and CTF-corrected image micrographs in RELION 3.1[55,56,68,69]. After manual picking, a first set of 65,131 segments were extracted in 1000-pixel boxes binned by 4 and a 10% inter-box distance. The tube diameter, rise and number of asymmetrical units were set to 125 Å, 4.75 Å and 21, respectively. Reference-free 2D classification was performed to select a single large class average for initial model generation with an estimated crossover distance of 700 Å. A second set of 381,233 smaller segments was extracted for the refinement applying a box size of 250 pixel with 10% inter-box distance and helical tube diameter, rise and asymmetrical unit values of 150 Å, 4.75 Å and 5, respectively. The initial model was re-scaled and re-windowed to match the un-binned particles and low-pass-filtered to 10 Å. 3D auto-refinement applying C1 symmetry, angular sampling, helical twist and rise values of 3.7°, 1.3° and 4.75 Å, respectively, yielded an ~4 Å resolution map. Imposing apparent C2 symmetry improved map resolution to 3.8 Å. Three out of four 3D classes summed up to 335,024 particles with similar cross-section densities. After additional steps comprising 3D class average selection, Bayesian polishing, CTF refinement and mask-generation, 65,122 particles were aligned by the final 3D auto-refinement with solvent-flattened FSC for map reconstruction. The final map was reconstructed with helical twist and rise values of 1.3° and 4.9 Å to an estimated resolution of 3.3 Å.

**Model building.** After map auto-sharpening in Phenix[70], the model was built de novo starting from a map region featuring an unusual backbone bulge with an associated bulky side-chain volume. The bulge was identified as $P_{66}GGAW_{70}$ in the LC-MS/MS-identified amino acid sequence. The model was built and refined in Coot, Chimera-Isolde as well as Phenix real-space refinement initially with and later without Amber gradients[71–75]. Molprobity validation[76] revealed model issues that were resolved by rebuilding of a single chain into the inverted map

with left-handed twist. Five 76-residue long chains in each protofilament were modeled and refined with non-crystallographic symmetry (NCS) restraints. In the final stages of refinement, we modeled Proline-66 as *cis*-isomer to fit the backbone carbonyl into the map, although a higher resolution is required to discriminate conclusively between *cis*- and *trans*-Proline. We also note additional densities close to residues Gly-65 and Ser-82, which may be occupied by ions or organic compounds of unknown origin. Phenix, Molprobity and EMDB validation[76–78] revealed map-model cross-correlation ($CC_{mask}$), EM-ringer and Molprobity-score values of 0.74, 5.1 and 1.4, indicative of a physically valid model with definite map support.

### Data analysis and visualization
Structures and derived data were analyzed and visualized using PyMol (Schrödinger, NY, USA) and Rstudio[79–83]. Molecular contact fingerprints, flexible structural alignments and buried surface areas as well as dissociation free energies of assemblies ($\Delta G_{diss}$) were obtained from Arpeggio, FATCAT and PISA webservers[84–86]. Solvation free energies ($\Delta G_{sol}$) were calculated in R following published procedures, using atomic solvent-accessible surface areas and H-bonds obtained by free-SASA and Arpeggio, respectively[31,85,87–89]. Sequences were aligned and visualized using Uniprot, Blast, ClustalOmega, ESPript and Jalview[90–94].

### Statistics and reproducibility
Histology and immunofluorescence staining protocols were optimized to stain tissue samples of 80 cats, as described previously[52]. Optimized protocols allowed us to obtain the images shown in Fig. 1 and Supplementary Fig. 1 at first trial ($n = 1$). The negative stain shown in Fig. 1b was selected from a collection of 13 micrographs comprising amyloid fibrils at different concentrations. The micrograph excerpt shown in Fig. 2a represents a single, straight amyloid selected from 15510 manually picked particles in 2652 micrographs (Supplementary Table 2). A selection of micrographs with variable amyloid particle distributions is shown in Supplementary Fig. 3a. LC-MS/MS data were collected four times on the same extract to optimize acquisition parameters for the final dataset ($n = 1$) (Supplementary Table 1).

## Data availability
The LC-MS/MS data have been deposited to the ProteomeXchange Consortium via the PRIDE[95] partner repository with the dataset id PXD035851. Cryo EM associated data have been deposited with id codes EMPIAR-11001, EMD-14726 and 7ZH7 at the EMPIAR, EMDB and RCSB data banks, respectively[77,96,97]. Other structures referenced in the manuscript are publicly available under the PDB accession codes, 4IP8, 6MST, 6DSO and 7ZH7. Raw histological and IF staining images are available on request to the corresponding author.

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

## Acknowledgements

This study was partially supported by Ricerca Corrente funding from Italian Ministry of Health to IRCCS Policlinico San Donato; by Centro di Ricerca Pediatrica, Fondazione Romeo and Enrica Invernizzi (Milan, Italy); Fondazione ARISLA (project TDP- 43-STRUCT); Italian Ministry of Research PRIN 2020 (20207XLJB2); AniCura Clinical Research Grant 2021. We acknowledge excellent support by Valter Fiore (Associazione La Cincia, Val Della Torre, Torino, Italy).

## Author contributions

Conceptualization and Supervision by A.C.S., F.L., M.B., E.Z., and S.R. Investigation and Analysis by A.C.S., T.S., G.Ma., V.S., F.L., F.F., C.P., M.M., P.M., M.N., and A.C.V. Funding acquisition and Resources by G.P., G.Me., M.B., S.F., E.Z., M.V. and S.R. Original draft by T.S. and S.R. Review and Editing by A.C.S., M.B., S.F. and E.Z. Data visualization by T.S., edited and reviewed by A.C.S., S.F. and S.R. Contribution to and approval of the submitted version by all authors.

## Competing interests

The authors declare no competing interests.
