## [Peer Review File · Nature Communications]

REVIEWER COMMENTS

Reviewer #1 (Remarks to the Author):

In this interesting and novel study, Rigano and colleagues report a new 3.3Å resolution cryo-EM structure of ex vivo amyloid fibrils from a cat with amyloid A (AA) amyloidosis. The fibril fold differs from those observed previously in ex vivo human and murine AA fibrils, thus contributing to the rapidly growing repertoire of amyloid folds. The authors report Pro cis-isomer in cat SAA (cSAA) fibrils, which has not been previously reported in amyloids. The study is well performed, technically sound and well written. Comments below relate mainly to the interpretation of results.

Pro in a Cys conformation in amyloid is an interesting observation. This Pro is 100% evolutionarily conserved in the SAA protein family and forms a tight turn between helices 2 and 3 in the native fold. Since Pro isomerization is a slow step in protein misfolding, the authors may consider whether the presence of Pro in a cis conformation in both native and fibrillary SAA might relate to the apparent ease of AA transmissibility in cats.

The authors should be careful relating fibril structure to the apparent prion-like transmissibility; the latter may be mediated by oligomeric intermediates (of unknown structure) rather than fibrils.

The fibrils were extracted from kidney, but the cat apparently also had liver amyloidosis. Is the fibril structure (or at least the gross fibril morphology) similar in the kidney and liver of this cat? If this analysis has not been performed, then the comparison between cat and cheetah amyloid fibril structure should be definitely toned down.

Based on the a.a. sequence similarity between cat and cheetah SAA, the authors propose similar AA amyloid structure in these animals. This is a stretch given the extreme sensitivity of the amyloid fold to the environmental conditions and conservative residue substitutions. The authors correctly state that the two a.a. substitutions that differentiate these feline species are compatible with the cat fibril structure reported here. They should probably stop at that given lack of further evidence.

Are there any extra densities seen in the EM map that cannot be accounted for by the protein? Such densities have been reported in mSAA and hSAA ex-vivo cryo-EM structures.

The authors may want to state from the start that their residue numbering is based on the pro-protein and includes an 18-residue propeptide.

In the Introduction the authors talk about SAA association with HDL and retinol and mention the evolutionarily conserved SAA function but do not tell their readers what it is. This primordial function was proposed to involve removal of cellular membrane debris and lipid degradation products from the sites of injury in inflammation.

Reviewer #2 (Remarks to the Author):

These comments exclude an evaluation of the quality/accuracy of cryo-EM data and interpretation.

In this manuscript the cryo-EM structure of cat AA amyloid protein is presented and compared with published structures of human and mouse AA amyloid proteins. The overriding theme is that cat AA amyloidosis develops and progresses via a prion-driven mechanism. Nowhere do the authors explain what they mean by prion-driven and prion capacity. A brief explanation needs to be provided in the Introduction. While horizontal transmission may be occurring, development of clinically significant AA amyloidosis ultimately requires sustained, cytokine-mediated, elevated production of SAA. The authors need to more adequately acknowledge the essential role of endogenous SAA, without which the SAA

seed and amyloid template introduced via horizontal transmission has no substrate upon which to act. That said, cat and cheetah SAA structures may be particularly effective amyloid templates and promote disease especially in the conditions of captivity.

Specific comments

Introduction.

Line 59. What are "light apolipoproteins"?

Lines 67- 68. This sentence could be re-worded. It is a bit dogmatic; while there are data supporting the idea that high levels of SAA lead to impaired lysosomal function and resultant accumulation of oligomers, this has not been conclusively proven.

Line 72. Add "A comparison of" cryo-EM structures....

Line 73. Cryo-EM was performed on ex vivo amyloid subunit proteins, not on amyloid deposits, which contain various other proteins and biomolecules.

Line 83. Define DHS at first usage.

Line 93. This is confusing. Does "deposited structure" refer to an amyloid deposit or something deposited in a database?

Results and Discussion, Figures, and Methods.

Figure S1A. Why not provide actual values rather than show data as a bar graph? If these data are to be presented, then state when blood was drawn - at necropsy versus prior to death. The data are probably not necessary.

Figure 1. The Congo red is fine, but the immunofluorescence is not informative. There is no mention in the Results, Methods or Figure Legend about ThT. It seems to be masking immunoreactivity. It would be more helpful to see anti-SAA staining in the absence of ThT and preferably via immunoperoxidase detection rather than immunofluorescence. Co-localization between Congo red and immunostaining could be revealed. In the legend, what is meant by "hotspots"?

The SAA antiserum is not well-described. A reference for or description of the virus-like particles needs to be provided. Also provide the cat SAA Uniprot ID used for designing the peptides, as well as the residue numbers which they represent (e.g., residues 10 – 20). Amyloid is notoriously sticky when it comes to binding antibodies. To make the staining more convincing, show an image in which primary antibody is left out.

Line 214. 1st sentence of Methods states that full details were presented in Reference 47. Reference 47 is in bioRxiv and is a preprint of this exact manuscript and provides no further information. This needs to be corrected.

A couple of other points:

1. SAA is a serum protein. The protein comprising SAA-derived amyloid is termed AA protein.
2. The numbering system used in the manuscript is bothersome. SAA, the protein found in the circulation, lacks a signal peptide; the N-terminus of secreted, mature SAA is most commonly denoted as residue 1, regardless of the species.

3. Show (in Supplementary Material) a denaturing gel electrophoresis picture (Coomassie blue-stained SDS-PAGE) of the preparation on which cryo-EM was performed.

4. Expand the description of the procedure used to solubilize the extracted fibrils. It simply states that they were dissolved in 8 M urea. At what temperature and for how long was this done; this is important since urea can induce chemical modifications. What is meant by "1/6 diluted" (% w/v or 1 part 8 M urea, 6 parts ammonium bicarbonate?).

5. Provide some LC-MS/MS data. Table S1 simply lists Uniprot entries. Show which peptides were detected and their relative abundance and/or show the mass of the intact AA protein. This is important because it is stated that the AA protein analyzed was 76 residues, yet there are no data supporting this.

Line 155. References for human and mouse structures are needed.

Reviewer #3 (Remarks to the Author):

The language could be a little simpler in places, e.g. "their molecular identities define specific disease forms and organ distribution" P4/L51 – does this mean 'different amyloid proteins aggregate in different places and cause different diseases'. A simpler more direct use of language might help the general reader.

P5/L51 – "highly polymorphic structures". I think this a slight misuse of language. I believe the authors mean that the mouse and human structures in Liberta et al 2019 are different – I agree. Saying that they are polymorphic suggests (to me) that each of the human and mouse samples were polymorphic (i.e. many different structures), which was not the case I believe.

P5/L86 – I think parenteral will not be well understood by the general reader without diving into the supporting literature. An extra few words of explanation would help the MS at this point.

Figure 1 seems a little superfluous to me.

The structural biology is generally fine, and I have only a few relatively minor questions on the MS or the presentation of the structural results.

1. I would ask the std cryoEM reviewer's question though. What is in the fibril segments that don't make it into their final reconstruction – they state the fibrils are homogeneous (P6/L118), but then only use 17% of their data – a sentence of two and a supp figure of their classification scheme would help here.

2. Re their point about the cis proline, I would guess they are correct, but I'm not 100% convinced by their figure at this resolution – a it is 'consistent with' a cis conformation but not definitive – as they themselves suggest in the methods. I think this is uncertain at this stage.

3. They state that the fibrils are left-handed but how was this determined? There's no evidence presented to support an experimental determination of this?

In general the MS is clear and well written, and presents a novel, interesting structure. However, I think it falls short of the standard for Nature Communications. The authors state that this is, 'the first ex vivo structure of a spontaneously occurring amyloid from an animal kept in a man-made habitat', which is true but of relatively minor impact in the field. The big interest would be the prion-like nature of the spread, but I cannot see anything in the structure or their description of it, that informs on the properties of this fibril that might contribute to that phenomenon. Without this, it is 'just' a solid piece of structural biology.

Overall therefore, I feel it falls short of the 'in field impact' required at Nature Comms.

Response to Reviewers

Responses are organized as follows:

#1) raised point

>>> response and explanation to reviewer

p. 5/ ll. 11: page / line numbers and relevant text changes/ highlighted in yellow in the text

Reviewer #1:

#1.1) In this interesting and novel study, Rigano and colleagues report a new 3.3Å resolution cryo-EM structure of ex vivo amyloid fibrils from a cat with amyloid A (AA) amyloidosis. The fibril fold differs from those observed previously in ex vivo human and murine AA fibrils, thus contributing to the rapidly growing repertoire of amyloid folds. The authors report Pro *cis*-isomer in cat SAA (cSAA) fibrils, which has not been previously reported in amyloids. The study is well performed, technically sound and well written. Comments below relate mainly to the interpretation of results.

>>> We are grateful to the positive evaluation by reviewer #1.

#1.2) Pro in a *cis* conformation in amyloid is an interesting observation. This Pro is 100% evolutionarily conserved in the SAA protein family and forms a tight turn between helices 2 and 3 in the native fold. Since Pro isomerization is a slow step in protein misfolding, the authors may consider whether the presence of Pro in a *cis* conformation in both native and fibrillary SAA might relate to the apparent ease of AA transmissibility in cats.

>>> Pro-66 adopts *trans* conformation in native lipid-free human SAA (PDB id: 4IP8) and retinol-bound mouse SAA (PDB id: 6PY0). However, the required unfolding during the α -to- β transition of native SAA to AA amyloid may allow for a spontaneous conversion from the *trans*- to the nearly isoenergetic *cis*-isomer, but could also be catalyzed by extracellular matrix (ECM) localized Peptidyl prolyl *cis/trans* isomerases (PPIases)¹⁻³. The *cis*-Pro-66 in feline AA amyloid orients the C-terminal meandering tail such that the side-chains of Pro-66, Ile-47, Trp-70 and Ala-72 form hydrophobic cluster/or steric zipper interactions. In murine and human AA amyloid the side-chains of both Pro-66 and Trp-70 point to the opposite side of the backbone, as visualized in Figure S3. We speculate that the *cis*-Proline induced re-orientation of the C-terminal meandering tail may allow feline AA amyloid to adopt a thermodynamically more stable structure, not accessible with *trans*-Pro-66.

#1.3) The authors should be careful relating fibril structure to the apparent prion-like transmissibility; the latter may be mediated by oligomeric intermediates (of unknown structure) rather than fibrils.

>>> We included specific information about non-fibrillar and fibrillar fragments of prions in the introduction. Regarding transmissibility of AA amyloidosis, we note that Zhang and co-workers⁴ have demonstrated that AA amyloid extracted from the feces and liver of cheetah worked as amyloid enhancing factor when administered to pre-inflamed mice. However, we have used careful language to indicate the uncertainties of our study, and of claims within the research field.

p. 5/ ll. 89: The classical prion disease transmissible spongiform ...

p. 11/ ll. 213: Whether auto-catalytic, self-perpetuating and ...

#1.4) The fibrils were extracted from kidney, but the cat apparently also had liver amyloidosis. Is the fibril structure (or at least the gross fibril morphology) similar in the kidney and liver of this cat? If this analysis has not been performed, then the comparison between cat and cheetah amyloid fibril structure should be definitely toned down. Based on the a.a. sequence similarity between cat and cheetah SAA, the authors propose similar AA amyloid structure in these animals. This is a stretch given the extreme sensitivity of the amyloid fold to the environmental conditions and conservative residue substitutions. The authors correctly state that the two a.a. substitutions that differentiate these feline species are compatible with the cat fibril structure reported here. They should probably stop at that given lack of further evidence.

>>> The reviewer has raised important points regarding polymorphism of amyloids across organs, and in this case between related species. We absolutely agree that these are important questions that we aim to address in future studies. At this stage we have decided to follow the reviewer's advice to tone down our speculative statement about similar or identical structures.

p 3/ ll. 47: The presented feline AA amyloid structure ...

p 10/ ll. 207: Remarkably, the amino acid sequence of AA amyloid extracted ...

#1.5) Are there any extra densities seen in the EM map that cannot be accounted for by the protein? Such densities have been reported in mSAA and hSAA ex-vivo cryo-EM structures.

>>> We noted additional densities close to residues Gly-65 and Ser-82, which may be occupied by ions or organic compounds of unknown origin. To visualize these additional

densities we have added the density cross-sections obtained from Relion in **Supplementary Figure 3B**.

Furthermore we added information in the text.

p. 14/ ll. 304: We also note additional densities ...

#1.6) The authors may want to state from the start that their residue numbering is based on the pro-protein and includes an 18-residue propeptide.

>>> added in main text and legends of **Figure 2 and 4**

p. 7/ ll. 125: ... corresponding to residue numbers 1-93 of the mature protein ...

p. 17/ ll. 343: Residues are numbered according to Uniprot entry Q1T770 ...

p. 19/ ll. 364: The amino acid sequences of hAA, ...

#1.7) In the Introduction the authors talk about SAA association with HDL and retinol and mention the evolutionarily conserved SAA function but do not tell their readers what it is. This primordial function was proposed to involve removal of cellular membrane debris and lipid degradation products from the sites of injury in inflammation.

>>> we thank the reviewer for pointing us to these interesting studies. We included two new references and changed the sentence accordingly.

p. 4/ ll. 62: This 12-14 kDa remarkably conserved apo-lipoprotein acts as ...

Reviewer #2:

#2.1) These comments exclude an evaluation of the quality/accuracy of cryo-EM data and interpretation. In this manuscript the cryo-EM structure of cat AA amyloid protein is presented and compared with published structures of human and mouse AA amyloid proteins.

>>> no comment required.

#2.2) The overriding theme is that cat AA amyloidosis develops and progresses via a prion-driven mechanism. Nowhere do the authors explain what they mean by prion-driven and prion capacity. A brief explanation needs to be provided in the Introduction.

>>> We agree that prion transmission has not been introduced properly which is crucial to put the structural findings in context.

p. 5/ ll. 89: The classical prion disease transmissible spongiform ...

#2.3) While horizontal transmission may be occurring, development of clinically significant AA amyloidosis ultimately requires sustained, cytokine-mediated, elevated production of SAA. The authors need to more adequately acknowledge the essential role of endogenous SAA, without which the SAA seed and amyloid template introduced via horizontal transmission has no substrate upon which to act. That said, cat and cheetah SAA structures may be particularly effective amyloid templates and promote disease especially in the conditions of captivity.

>>> The essential role of increased elevated endogenous SAA levels is described in several instances,

p. 4/ ll. 69: Chronic inflammation leads to ...

p. 5/ ll. 97: ... accelerated AA amyloidosis in chronically inflamed animals ...

p. 11/ ll. 216: ... demonstrated to infect inflamed mice ...

p. 11/ ll. 220: ... in crowded populations of chronically inflamed felids ...

#2.4) Specific comments to Introduction

#2.4.1) # Line 59. What are “light apolipoproteins”?

>>> we deleted “light” as the molecular weight is given in the same sentence

p. 4/ ll. 62: This 12-14 kDa remarkably ...

#2.4.2) Lines 67- 68. This sentence could be re-worded. It is a bit dogmatic; while there are data supporting the idea that high levels of SAA lead to impaired lysosomal function and resultant accumulation of oligomers, this has not been conclusively proven.

>>> added “reportedly” to the phrase

p. 4/ ll. 70: ... to an extent that macrophages reportedly fail to ...

#2.4.3) # Line 72. A comparison of cryo-electron microscopy (EM) structures of ...

and #Line 74: Cryo-EM was performed on ex vivo amyloid subunit proteins, not on amyloid deposits, which contain various other proteins and biomolecules.

>>> re-phrased

p. 5/ ll. 75: A comparison of cryo-electron microscopy (EM) structures of ...

#2.4.4) Line 83. Define DHS at first usage.

>>> definition added

p. 5/ ll. 85: Among domestic short hair (DSH) cats,

#2.4.5) Line 93. This is confusing. Does “deposited structure” refer to an amyloid deposit or something deposited in a database?

>>> re-phrased

p. 6/ ll. 107: ... but adopts a unique fold.

#2.5) specific to Results and Discussion, Figures, and Methods.

#2.5.1) Figure S1A. Why not provide actual values rather than show data as a bar graph?

If these data are to be presented, then state when blood was drawn - at necropsy versus prior to death. The data are probably not necessary.

>>> added to the legend of *Supplementary Figure 1*

#2.5.2) Figure 1. The Congo red is fine, but the immunofluorescence is not informative.

There is no mention in the Results, Methods of Figure Legend about ThT. It seems to be masking immunoreactivity. It would be more helpful to see anti-SAA staining in the absence of ThT and preferably via immunoperoxidase detection rather than immunofluorescence. Co-localization between Congo red and immunostaining could be revealed. To make the staining more convincing, show an image in which primary antibody is left out.

>>> We added missing information about the Thioflavin S staining in materials&methods, and the legend of Figure S1, as well as in the Results section. We removed the term hotspots. Herein we also refer to the more detailed M&Ms of the related manuscript.

p. 7/ ll. 123: ... of Thioflavin-positive amyloids in all three organs

p.12/ ll. 237: ... were stained with 1% (w/V) Thioflavine S and DAPI, respectively.

>>> We also thank the reviewer for pointing out apparent masking effects in the composite image, not revealing any co-localization. However, the lack of co-localization was mainly due to non-optimal presentation of our existing imaging data. To facilitate interpretation of the staining, we added images of the separately acquired Thioflavin S (green) and anti-SAA (magenta) stainings as separate panels in *Supplementary Figure 1*. Furthermore, we noticed non-matching relative intensity profiles in the two channels of the digitally merged composite image, which have been carefully matched in the new version. Furthermore we substituted red by magenta to generate color-blind friendly figures. In the new composite images, co-localized staining at matched relative signal intensity appears white. Attached to this report we also provide the raw grey-scale 16 bit images of the separate channels, as well as secondary antibody control images that were generated for organs of other cats presented in our second study (*Attachments 1 and 2*).

#2.5.4) In the legend of Figure S1, what is meant by “hotspots”?

>>> Indeed, this term is misleading and was deleted.

#2.5.5) The SAA antiserum is not well described. A reference for or description of the virus-like particles needs to be provided. Also provide the cat SAA Uniprot ID used for

designing the peptides, as well as the residue numbers which they represent (e.g., residues 10 – 20). Amyloid is notoriously sticky when it comes to binding antibodies.

>>> We have added the Uniprot ID, added a new reference for the virus-like particles, and also refer to our related study.

p.12 / ll. 238: For immunofluorescence, serum was ...

#2.5.6) Line 214. 1st sentence of Methods states that full details were presented in Reference 47. Reference 47 is in bioRxiv and is a preprint of this exact manuscript and provides no further information. This needs to be corrected.

>>> corrected

#2.6) a couple of other points:

#2.6.1) SAA is a serum protein. The protein comprising SAA-derived amyloid is termed AA protein.

>>> We believe to have followed this definition except for the paragraph “Cat’s distinct AA amyloid structure buries its unique eight-residue insert between the two proto-filaments and is predicted as the most stable assembly”. We have changed the abbreviation of the human, mouse and cat AA amyloid structures to hAA, mAA and cAA.

p. 9/ ll. 172: Compared to the 54- residue ...

#2.6.2) The numbering system used in the manuscript is bothersome. SAA, the protein found in the circulation, lacks a signal peptide; the N-terminus of secreted, mature SAA is most commonly denoted as residue 1, regardless of the species.

>>> The reviewer has pointed to a general problem that residue numbers of many PDB-deposited models do not match their Uniprot-referenced sequences. A new re-numbering website (<http://dunbrack.fccc.edu/PDBrenum/>) was created by Faezov and co-workers ⁵ to address that “inconsistent numbering schemes compromise structural bioinformatics studies that seek to compare multiple structures of a single protein or structures within protein families across the PDB.” Our numbering system matches its referenced Uniprot entry, in contrast to many other structures, i.e. human and mouse AA amyloid structures (with PDB codes 6MST and 6DSO). *To help the reader, we have also included the specific information about the cleaved signal peptide in the main text and the Figure legends at several instances.*

p. 7/ ll. 125: ... corresponding to residue numbers 1-93 of the mature protein ...

p. 17/ ll. 343: Residues are numbered according to Uniprot entry Q1T770, ...

p. 19/ ll. 364: The amino acid sequences of ...

#2.6.3) Show (in Supplementary Material) a denaturing gel electrophoresis picture (Coomassie blue-stained SDS-PAGE) of the preparation on which cryo-EM was performed.

>>> added to Supplementary Figure 1D

#2.6.4) Expand the description of the procedure used to solubilize the extracted fibrils. It

simply states that they were dissolved in 8 M urea. At what temperature and for how long was this done; this is important since urea can induce chemical modifications. What is meant by “1/6 diluted” (% w/v or 1 part 8 M urea, 6 parts ammonium bicarbonate?).

>>> corrected

p. 13/ ll. 256: Fibril proteins were solubilized for 1 hour at room temperature ...

p. 13/ ll. 260: ... to a final urea concentration of 1.3 M, ...

#2.6.5) Provide some LC-MS/MS data. Table S1 simply lists Uniprot entries. Show which peptides were detected and their relative abundance and/or show the mass of the intact AA protein. This is important because it is stated that the AA protein analyzed was 76 residues, yet there are no data supporting this.

>>> We added Supplementary Figure 2 revealing the relative abundance of peptides detected for Uniprot entry Q1T770. The data reveal SAA residues 19-111 as relevant fragment, and refer to this in the results at following lines.

p. 7/ ll. 124: Peptides covering residue numbers ...

p.8/ ll. 146: Residues 95-111, identified by LC-MS/MS ...

>>> We have deposited the LC-MS/MS data to the PRIDE repository with the dataset id PXD035851 (see also Supplementary Table 1). The reviewer may access the deposited data: **Username:** reviewer_pxd035851@ebi.ac.uk, **Password:** OnCWq1fK

#2.6.5) #Line 155. References for human and mouse structures are needed.

>>> fixed

p.9/ ll. 170: Although the amino acid sequences of the published human and mouse

Reviewer #3

#3.1) The language could be a little simpler in places, e.g. “their molecular identities define specific disease forms and organ distribution” P4/L51 – does this mean ‘different amyloid proteins aggregate in different places and cause different diseases’. A simpler more direct use of language might help the general reader.

>>> re-phrased

p. 4/ ll. 54: Each of more than 50 disease-causing amyloidogenic proteins ...

#3.2) # P5/L51 – “highly polymorphic structures”. I think this a slight misuse of language. I believe the authors mean that the mouse and human structures in Liberta et al 2019 are different – I agree. Saying that they are polymorphic suggests (to me) that each of the human and mouse samples were polymorphic (i.e. many different structures), which was not the case I believe.

>>> re-phrased

p. 5/ ll. 77: ...amyloid, but distinct structures despite 76% sequence ...

#3.3) # P5/L86 – I think parenteral will not be well understood by the general reader without diving into the supporting literature. An extra few words of explanation would help the MS at this point.

>>> re-phrased

p. 5/ ll. 97: ... after intravenous injection or oral administration of amyloid ...

#3.4) Figure 1 seems a little superfluous to me.

>>> We have chosen to present the complete case study from disease diagnosis to molecular structure. We believe it may be of interest for readers to have a full view on these amyloids from the tissue of origin to the structure.

#3.5) The structural biology is generally fine, and I have only a few relatively minor questions on the MS or the presentation of the structural results.

>>> appreciated

#3.5.1) I would ask the std cryoEM reviewer's question though. What is in the fibril segments that don't make it into their final reconstruction – they state the fibrils are homogeneous (P6/L118), but then only use 17% of their data – a sentence of two and a supp figure of their classification scheme would help here.

>>> A percentage of 17% for the extracted segments in the final reconstruction is not extraordinary. Below we list initially picked versus finally used segments from two *ex vivo* structures that have been cited in our study, both structures representing dominant fibril morphologies:

Structure	ini	final	[%]	morphology	Publication
cat AA amyloid	381,233	65,122	17	Dominant (not quantified)	This study
mouse AA amy, EMD-8910	137,956	21,024	15	Dominant (> 90%)	Liberta, F. et al. Nat Commun 10, 1104 (2019).
infectious prion (263K strain)	337,368	15,884	4	Dominant (not quantified)	Kraus, A. et al. Molecular Cell 81, 4540-4551.e6 (2021).

Herein, we provide further information on the initial 3D classification after the first 3D auto-refinement with particle numbers of each class:

Evidently, three initially “good” looking classes sum up to 335k particles, thus about 88% of the initially picked particles. The following iterative Bayesian reconstruction selects particle subsets based on optimal particle alignment, for best resolution. The exact molecular features leading to de-selection of a large number of particles (that seem visually similar in the initial 3D classes) are not entirely clear to us. There might be slight conformational variations and alterations in twist and distance that precluded the other particles from the presented reconstruction. However, even the kink comprising the modeled *cis*-Proline is recognizable in all three classes. We support open data policies and have deposited the raw micrographs under EMPIAR-11001, thus anyone interested can analyze the dataset further. However, apart from lower resolution maps for additional subsets present in the dataset, we do not expect any deviations that would alter the main conclusions of our study. We added the relevant information about the 3D classes to Supplementary Table 2 and in the main text.

p. 14/ ll. 288: Three out of four 3D classes summed up to 335,024 particles with similar cross-section densities.

#3.5.2) Re their point about the *cis* proline, I would guess they are correct, but I'm not 100% convinced by their figure at this resolution – as it is 'consistent with' a *cis* conformation but not definitive – as they themselves suggest in the methods. I think this is uncertain at this stage.

>>> In our opinion we made this point very clear in the text. We have built the best possible model, and in our opinion, the model comprising *cis*-Proline fits better to the map, and is also in line with the observation that the side-chains of following residues in the other two available structures point to opposite directions (Supplementary Figure 3).

p. 8/ ll. 150: ... segment comprising Pro-66 *modeled* as *cis*-isomer ...

p. 14/ ll. 302: ... we modeled Proline-66 as *cis*-isomer to fit the backbone carbonyl into the map, although a higher resolution is required to discriminate conclusively between *cis*- and *trans*-Proline.

#3.5.3) They state that the fibrils are left-handed but how was this determined? There's no evidence presented to support an experimental determination of this?

>>> The initial map was actually right-hand twisted, but we failed to build a model with good statistics into this map. We could fix these issues by building the model into the inverted map. We also note that the vast majority of amyloids is left-handed. Therefore we concluded that this particular amyloid is very likely left-handed. The information about the model re-building into the inverted map was already described in our previous version, and is now found here:

p. 14/ ll. 299: Molprobitry validation revealed model issues that were resolved by rebuilding of a single chain into the inverted map with left-handed twist.

#3.5.4) In general the MS is clear and well written, and presents a novel, interesting structure.

>>> We thank the Referee for this comment.

#3.5.5) However, I think it falls short of the standard for Nature Communications. The authors state that this is, ‘the first *ex vivo* structure of a spontaneously occurring amyloid from an animal kept in a man-made habitat’, which is true but of relatively minor impact in the field. The big interest would be the prion-like nature of the spread, but I cannot see anything in the structure or their description of it, that informs on the properties of this fibril that might contribute to that phenomenon. Without this, it is ‘just’ a solid piece of structural biology. Overall therefore, I feel it falls short of the ‘in field impact’ required at Nature Comms.

>>> Obviously, we disagree. The presented novel feline AA amyloid fold comes from a very specific environment in which AA amyloidosis, a rare disease, is strikingly common. The data and analysis shown in this manuscript also helps the field to contextualize previous observations of prion-like transmission in captive cheetah. In particular, we have put forward novel testable hypotheses about increased prion propensity of the feline SAA variant. Inclusion of the feline-specific eight-residue insert in the amyloid core increases amyloid core mass and stability (Supplementary Figure 9, also comprising additional solvation free energy estimates), which likely contributes to increased environmental persistence and clearance resistance in the host. We believe that our data will inspire other researchers in the field to design new experiments to test our hypothesis.

p. 3/ ll. 46: Inclusion of an eight-residue insert unique ...

p. 10/ ll. 195: Structure-based solvation free energy (ΔG_{sol}) calculations ...

p. 15/ ll. 314: Solvation free energies (ΔG_{sol})

We also want to comment on the prion-related statement that the reviewer ‘cannot see anything in the structure or their description of it, that informs on the properties of this fibril that might contribute to that phenomenon.

>>> We have clarified the discriminative features of prion versus amyloid throughout the manuscript, and highlight following definition:

p. 11/ ll. 213: Whether auto-catalytic, self-perpetuating and self-propagating amyloids are defined as prions, depends primarily on their capacity to undergo the full transmission cycle of an infective agent.

>>> Thus, structural features can only contribute, but never be sufficient to explain prion-like diseases. Instead, certain structural features may enhance the propensity of an amyloid to be transmitted like a prion. *Is there evidence for a prion-like disease spread in cat shelters?* We don’t have direct evidence, but clear indications: (i) extreme AA amyloidosis prevalence that is unlikely to be explained by chronic inflammation alone (ii) identification of SAA in cat bile indicates potential feces-oral transmission (iii) the same phenomenon is observed in captive cheetah with almost identical AA amyloid amino acid sequence, and with reported prion-like disease transmission. *Do we reveal any structural features that might contribute an increased prion propensity?* First of all we should state that no specific structural motifs are associated with prion-like properties. Conversely, most amyloids, thus also the human and mouse variants, might principally be associated with prion-like disease transmission under certain environmental and host

conditions. Hence, we only emphasize that increased mass and stability of the feline amyloid variant could contribute to enhance transmissibility and thus infectivity.

References

1. Lin, W. *et al.* Peptidyl prolyl cis/trans isomerase activity on the cell surface correlates with extracellular matrix development. *Commun. Biol.* **2**, 1–11 (2019).
2. Lu, K. P., Finn, G., Lee, T. H. & Nicholson, L. K. Prolyl cis-trans isomerization as a molecular timer. *Nat. Chem. Biol.* **3**, 619–629 (2007).
3. Wedemeyer, W. J., Welker, E. & Scheraga, H. A. Proline Cis–Trans Isomerization and Protein Folding. *Biochemistry* **41**, 14637–14644 (2002).
4. Zhang, B. *et al.* Fecal transmission of AA amyloidosis in the cheetah contributes to high incidence of disease. *Proc. Natl. Acad. Sci. U. S. A.* **105**, 7263–7268 (2008).
5. Faezov, B. & Dunbrack, R. L. PDBrenum: A webserver and program providing Protein Data Bank files renumbered according to their UniProt sequences. *PLoS ONE* **16**, e0253411 (2021).
6. Davis, I. W. *et al.* MolProbity: all-atom contacts and structure validation for proteins and nucleic acids. *Nucleic Acids Res.* **35**, W375–383 (2007).
7. Sawaya, M. R., Hughes, M. P., Rodriguez, J. A., Riek, R. & Eisenberg, D. S. The expanding amyloid family: Structure, stability, function, and pathogenesis. *Cell* **184**, 4857–4873 (2021).

DAPI

Thioflavin

anti-SAA

kidney

spleen

liver

Dapi

ThioS

+/- sera + 2nd AB

Spleen AI265

Spleen AG950

Attachment 2.1

Dapi

ThioS

+/- sera + 2nd AB

Kidney AI263

Kidney AG953

Attachment 2.2

REVIEWER COMMENTS

Reviewer #2 (Remarks to the Author):

As noted in the original review, the comments of this reviewer are not an evaluation of the cryo EM results or their structural interpretation.

Specific comments:

Figure 1. The histology showing Congo red staining is fine and should be included; it confirms the presence of amyloid. The immunofluorescence and thioflavin S staining are no more informative than they were in the original submission. There is no control to verify antiserum specificity or to demonstrate the extent of background thioflavin S staining in amyloid-negative tissue. The authors could simply omit Figure 1B. Keep the focus on structural biology.

Supplementary Figure 1. Panel a is not relevant, and along with panels b and c, could be deleted. Keep panel d. The legend to panel d needs to indicate the detection method used to visualize bands in gel – Coomassie blue or immunodetection?

Supplementary Figure 2. This figure and its legend are very difficult to understand. It needs to be simplified. Why not present the cat sequence from the database (1 sequence is sufficient), below that show the peptides specifically identified in their LC-MS/MS analysis, below that show some indication of their relative abundance in the analysis, and below that indicate those identified in the fibril by cryoEM.

The fibril cryo EM indicated the fibril comprised 76 residues (19 – 94) or 1- 76 of the mature protein present in serum. The authors need to address the presence of the peptides identified by LC-MS/MS that are not part of the fibril. If they have achieved a clean fibril preparation, how do they account for the extra C-terminal SAA peptides present in the prep? The SDS-PAGE looks exceptionally clean.

Supplementary Table 1. I looked up the PRIDE dataset id listed in this table = PXD035851 and the search gave no results. Please check that the correct id is listed.

General comments:

In addressing the issues raised in the first reviews, the authors have packed in more references and figures, but have not really enhanced the paper. It seems that because the paper has been submitted to an online-only journal, there has been less regard to maintain succinctness. At times the text is rambling, compromising the focus of the manuscript. The 111 references, some of which date back to 1972, are excessive in number and duplicative; in addition, reference formatting is inconsistent, and in some cases (e.g., the journal *Amyloid*), the name of the journal is incorrectly noted. While this may not be a substantive point, it is annoying as is the repeated use of the "amyloids" instead of simply amyloid and the use of "cat's SAA and mouse's SAA" instead of cat and mouse SAA. Somewhere in the middle of the Results, the authors start using the abbreviations cSAA, mSAA and hSAA for cat, mouse and human SAAs - why not use the abbreviations at the outset? These issues exemplify a pervasive lack of attention.

REVIEWER COMMENTS

Reviewer #2 (Remarks to the Author):

As noted in the original review, the comments of this reviewer are not an evaluation of the cryo EM results or their structural interpretation.

Specific comments:

#1.1) Figure 1. The histology showing Congo red staining is fine and should be included; it confirms the presence of amyloid. The immunofluorescence and thioflavin S staining are no more informative than they were in the original submission. There is no control to verify antiserum specificity or to demonstrate the extent of background thioflavin S staining in amyloid-negative tissue. The authors could simply omit Figure 1B. Keep the focus on structural biology.

Supplementary Figure 1. Panel a is not relevant, and along with panels b and c, could be deleted. Keep panel d. The legend to panel d needs to indicate the detection method used to visualize bands in gel – Coomassie blue or immunodetection?

>>> Herein, we address the two related points. We have removed Figure 1B from the main manuscript, but kept the staining figures as supplementary material. We have also shifted the SDS-PAGE image to Supplementary Fig. 2.

We emphasize that the presented images were taken from a tissue collection obtained from 79 cats (Ferri, F. *et al. bioRxiv* **2022.05.04.490646**). AA amyloidosis was identified in at least one organ of 48 cats by Congo red and immunofluorescence (IF) staining. However, organs of 13 cats were not stained positive by Congo red or IF. For purely reviewing purposes, we have attached the stain of an amyloid-free tissue at the end of this letter, in direct comparison to a positively stained tissue. These (or similar images) will be added to the second manuscript that is currently under review in the journal *Veterinary Pathology*.

In general, we disagree with Reviewer 2: in *ex vivo* structure biology we think it is relevant to provide some information on the tissue from which the samples are extracted.

p. 7 / ll. 119 modified the paragraph to highlight that this specific cat represents a case study from a collection of 79 cats, of which 35 cats had amyloid deposits in the liver, kidney and spleen.

#1.2) Supplementary Figure 2. This figure and its legend are very difficult to understand. It needs to be simplified. Why not present the cat sequence from the database (1 sequence is sufficient), below that show the peptides specifically identified in their LC-MS/MS analysis, below that show some indication of their relative abundance in the analysis, and below that indicate those identified in the fibril by cryoEM.

>>> We modified Supplementary Figure 2 accordingly.

#1.3) The fibril cryo EM indicated the fibril comprised 76 residues (19 – 94) or 1- 76 of the mature protein present in serum. The authors need to address the presence of the peptides identified by LC-MS/MS that are not part of the fibril. If they have achieved a clean fibril preparation, how do they account for the extra C-terminal SAA peptides present in the prep? The SDS-PAGE looks exceptionally clean.

>>> We have already commented on this observation in our previous version, now to be found on

p. 8 / ll. 145 Residues 95-111, identified by LC-MS/MS as component of the extracted AA amyloid, were not visible in the map.

We can only speculate that the “missing” residues are simply too flexible/disordered to be observed in the map. It is very common that part of the polypeptide is invisible in Cryo-EM structures of amyloid fibrils.

#1.3) Supplementary Table 1. I looked up the PRIDE dataset id listed in this table = PXD035851 and the search gave no results. Please check that the correct id is listed.

>>> The correct id is listed. The dataset has not officially been released. As we have already written in our last response, the reviewer can access the data using following login details:

website: <https://www.ebi.ac.uk/pride/login>

Username: reviewer_pxd035851@ebi.ac.uk

Password: OnCWq1fK

#1.4) General comments:

In addressing the issues raised in the first reviews, the authors have packed in more references and figures, but have not really enhanced the paper. It seems that because the paper has been submitted to an online-only journal, there has been less regard to maintain succinctness. At times the text is rambling, compromising the focus of the manuscript. The 111 references, some of which date back to 1972, are excessive in number and duplicative; in addition, reference formatting is inconsistent, and in some cases (e.g., the journal *Amyloid*), the name of the journal is incorrectly noted. While this may not be a substantive point, it is annoying as is the repeated use of the “amyloids” instead of simply amyloid and the use of “cat’s SAA and mouse’s SAA” instead of cat and mouse SAA. Somewhere in the middle of the Results, the authors start using the abbreviations cSAA, mSAA and hSAA for cat, mouse and human SAAs - why not use the abbreviations at the outset? These issues exemplify a pervasive lack of attention.

>>> We strongly disagree with the overall assessment by Referee 2 but for sake of brevity here we list only the modifications and amendments compatible with the above comments.

We have removed some “duplicative” citations to stay below 100 references in total. While about 55-60 of these citations are related to the scientific background, the additional ~40 citations are referring to methodological/technical work essential for us

to obtain and analyze our work, therefore deserving credit. We have also removed the abbreviations, since they were only used in a single paragraph.

Figure 1. Negative control staining.

DAPI, thioflavin S (ThioS) and immune serum staining are shown for AA amyloid positive and negative tissue samples obtained from the spleen and parotid tissue of cats AG573 and AI246, respectively.